# Mendelian Randomization Reveals: Triglycerides and Sensorineural Hearing Loss

**DOI:** 10.3390/bioengineering11050438

**Published:** 2024-04-29

**Authors:** Shun Ding, Yixuan Liu, Tingting Duan, Peng Fang, Qiling Tong, Huawei Li, Huiqian Yu

**Affiliations:** 1ENT Institute and Department of Otorhinolaryngology, Eye and ENT Hospital, State Key Laboratory of Medical Neurobiology and MOE Frontiers Center for Brain Science, Fudan University, 83 Fenyang Road, Shanghai 200031, China; dingshunent@fudan.edu.cn (S.D.); 22111260031@m.fudan.edu.cn (Y.L.); 23211260026@m.fudan.edu.cn (Q.T.); 2NHC Key Laboratory of Hearing Medicine, Fudan University, Shanghai 200031, China; 3Department of Otolaryngology, Head and Neck Surgery, The First Affiliated Hospital, Hainan Medical University, Haikou 570102, China; denieceduan@163.com; 4Department of Orthopedics, Nanjing Jinling Hospital, Affiliated Hospital of Medical School, Nanjing University, Nanjing 210002, China; njumasterfangpeng@163.com

**Keywords:** triglycerides, sensorineural hearing loss, mendelian randomization, meta-analysis

## Abstract

Background: Sensorineural hearing loss (SNHL) is a multifactorial disorder with potential links to various physiological systems, including the cardiovascular system via blood lipid levels such as triglycerides (TG). This study investigates the causal relationship between TG levels and SNHL using Mendelian randomization (MR), which offers a method to reduce confounding and reverse causality by using genetic variants as instrumental variables. Methods: Utilizing publicly available genome-wide association study (GWAS) data, we performed a two-sample MR analysis. The initial analysis unveiled a causal relationship between TG (GWAS ID: ebi-a-GCST90018975) and SNHL (GWAS ID: finn b-H8_HL_SEN-NAS). Subsequent analysis validated this through MR with a larger sample size for TG (GWAS ID: ieu-b-111) and SNHL. To conduct the MR analysis, we utilized several methods including inverse-variance weighted (IVW), MR Egger, weighted median, and weighted mode. We also employed Cochrane’s Q test to identify any heterogeneity in the MR results. To detect horizontal pleiotropy, we conducted the MR-Egger intercept test and MR pleiotropy residual sum and outliers (MR-PRESSO) test. We performed a leave-one-out analysis to assess the sensitivity of this association. Finally, a meta-analysis of the MR results was undertaken. Results: Our study found a significant positive correlation between TG and SNHL, with OR values of 1.14 (95% CI: 1.07–1.23, *p* < 0.001) in the IVW analysis and 1.09 (95% CI: 1.03–1.16, *p* < 0.006) in the replicate analysis. We also found no evidence of horizontal pleiotropy or heterogeneity between the genetic variants (*p* > 0.05), and a leave-one-out test confirmed the stability and robustness of this association. The meta-analysis combining the initial and replicate analyses showed a significant causal effect with OR values of 1.11 (95% CI: 1.06–1.16, *p* = 0.01). Conclusion: These findings indicate TG as a risk factor for SNHL, suggesting potential pathways for prevention and intervention in populations at risk. This conclusion underscores the importance of managing TG levels as a strategy to mitigate the risk of developing SNHL.

## 1. Background

Globally, an estimated 1.5 billion individuals suffer from hearing impairment. Among them, around 430 million manifest moderate to severe auditory deficiencies in their superior ear. The predominant category of this auditory affliction is sensorineural hearing loss [1]. Sensorineural hearing loss (SNHL) arises from organic damage to the hair cells, stria vascularis, spiral ganglia, auditory nerve, or central auditory pathways. The etiological factors underpinning sensorineural deafness are multifaceted, encompassing sudden onset deafness, ototoxicity, presbycusis, noise-induced hearing loss, infections of the auditory system, and autoimmune inner ear diseases. The pathogenesis is potentially associated with inflammatory reactions, cochlear microcirculatory disturbances, and diminished afferent signaling from hair cells [2]. Among these, damage to the hair cells stands as the principal factor underlying acquired SNHL. Owing to their non-regenerative capability, injury to hair cells frequently results in irreversible hearing impairment [3]. Hence, prompt detection, prevention, and intervention for SNHL are imperative.

Triglycerides (TG) are hydrophobic and require plasma lipoproteins for transportation to various tissues in the body. Circulating TG concentrations fluctuate based on dietary intake, inherent genetic and metabolic attributes, and the intricate hormonal interplay between internal organs [4]. The recent global surge in hyperlipidemia and its concomitant disorders can be attributed to shifts in lifestyle and dietary patterns. TG, a pivotal constituent of serum lipids, have been implicated in the initiation and progression of various cardiovascular pathologies [5,6,7]. Yet, beyond their impact on the cardiovascular system, TG may exert influence on other physiological systems. Studies suggest that individuals experiencing sudden sensorineural hearing loss (SSNHL) frequently present with heightened triglyceride concentrations, correlating with a less promising prognosis [8,9]. Evans MB et al. demonstrated an increased risk of hearing loss with elevated serum triglycerides [10]. Nonetheless, research has established that, when accounting for age and gender, dyslipidemia does not appear to be linked with SNHL [11]. Chang IJ and colleagues [12] neither identified a clear association between TG and SSNHL nor conclusively dismissed such a linkage. Further investigations are imperative to elucidate the potential correlation, or absence thereof, between serum lipids and SSNHL. Hence, delving into the intricate interplay between TG and SNHL is vital, as understanding this nexus could aid in reducing the incidence of SNHL within this specific patient population.

There are still many unresolved questions regarding the relationship between TG levels and hearing loss. Although some studies have suggested that high TG levels may be associated with hearing decline, the exact mechanisms of this connection remain unclear. It is not known whether the impact is direct (TG directly affecting the inner ear structures [13]) or through other indirect pathways (such as inflammation, vascular health [14], etc.). Moreover, most existing studies are observational and cannot establish a causal relationship [15]. TG are affected by many confounding factors. For example, TG may cause hearing loss through affecting microvascular changes, but other confounding variables such as otitis media [16], rhinitis [17], hypertension [18], and sleep apnea [19] can also impact the results. These factors each influence ear microvascular health and functionality through different mechanisms, potentially masking or exaggerating the effects of TG on hearing. Therefore, controlling these variables in studies assessing the relationship between TG and hearing loss is crucial, as it helps to reveal a clearer and more accurate causal connection.

Mendelian randomization (MR) using genetic variants as instrumental variables effectively controls confounding factors and strengthens the reliability of causal inference, marking a significant advantage over traditional observational studies [20]. Observational research is often limited by its inability to fully isolate all potential confounding variables, which may lead to biased interpretations of causal relationships [21]. This study, by simulating the conditions of randomized controlled trials, utilizes MR to reveal the complex relationship between TG and SNHL, providing a more precise and scientific methodology that helps researchers more accurately assess the effects of health interventions.

## 2. Methods

### 2.1. Data Sources and Study Design

This study utilizes extensive GWAS summary datasets in which all participants provided informed consent during their original studies. Since our analysis relied exclusively on summary-level statistics, no further ethical approval was required. The GWAS data employed in the initial MR analysis of triglycerides originated from the White Europe meta-analysis, featuring a sample size of 343,992 cases and bearing the GWAS ID “ebi-a-GCST90018975.” For the repeated MR analysis of triglycerides, we again utilized data from the White Europe meta-analysis, encompassing a sample size of 441,016 cases and identified by the GWAS ID “ieu-b-111.” The GWAS datasets were accessed through the following website: https://gwas.mrcieu.ac.uk/ (accessed on 15 January 2024). The summary data for SNHL was retrieved from the FinnGen Biobank, comprising 15,952 cases and 196,592 European bloodline controls within the dataset. This data is affiliated with the IEU Open GWAS project, identified by the GWAS ID “finn b-H8_HL_SEN-NAS”. For more detailed information, kindly refer to Appendix A.

We have conducted a two-way, two-sample Mendelian randomization (MR) study to investigate the causal link between TG and SNHL, employing single nucleotide polymorphisms (SNPs) as instrumental variables (IVs). These SNPs needed to satisfy three critical assumptions [22,23,24,25] (Figure 1). Assumption 1: These SNPs must exhibit a strong correlation with the exposure. Assumption 2: These SNPs should affect the outcome solely through the exposure. Assumption 3: These SNPs should not be associated with confounding factors. Our analytical process encompassed six key components: (1) Procurement of GWAS data for both TG and SNHL. (2) Prudent selection of the most suitable SNPs as instrumental variables for TG. (3) Extraction of SNPs from the chosen instrumental variables and integrating them into SNHL GWAS data. (4) Standardization of TG and SNHL GWAS data to ensure uniformity. (5) Execution of MR analysis, along with heterogeneity, pleiotropy, and sensitivity assessments. (6) Replication of MR analysis for validation, followed by meta-analysis of MR results. Figure 2 illustrates the entire analytical workflow.

### 2.2. IVs

We exclusively used summary-level data from GWAS for both TG and SNHL, ensuring that all participants in these datasets had provided informed consent. To establish criteria for IVs selection in the exposed group data, a significance threshold of *p* < 5 × 10^−8^ was set [26]. To mitigate bias arising from linkage disequilibrium (LD) in the analysis, this study required LD conditions for SNPs closely associated with TG to meet the criteria of r^2^ < 0.001 [27], with a genetic distance of 10,000 kb [28]. Relevant information, including chromosome (CHR), effect allele (EA), non-effect allele (NEA), the effect allele frequency (EAF), effect size (ES or β), standard error (S), and *p*-value, was extracted. To prevent weak IVs bias, an F-statistic greater than 10 was defined as indicating the absence of weak IVs bias [29].

### 2.3. Statistical Analysis

A two-sample MR analysis to explore the causal relationship between TG and SNHL was conducted using the ‘TwoSampleMR’ package within R software (version 4.1.2). Various MR methods were employed, including IVW (inverse variance weighted), MR-Egger, weighted median, and weighted mode. The primary approach was the random-effects IVW method, with MR-Egger, weighted median, and weighted mode serving as secondary methods to ensure result robustness.

To evaluate result heterogeneity, we employed the I^2^ index and Cochran’s Q statistic in the MR-IVW analysis, and Rucker’s Q statistic in the MR-Egger analysis, with *p* > 0.05 indicating the absence of heterogeneity. Horizontal multiple effects were assessed using both the MR-Egger and MR-PRESSO methods, with *p* > 0.05 indicating no horizontal multiple effects.

We utilized MR-PRESSO and Radial MR to detect outlier SNPs, and after removing these SNPs, the MR results were re-estimated to ensure the robustness of the findings. Additionally, leave-one-out analyses were performed to investigate whether individual SNPs had a notable impact on the causal relationship between exposure and outcome.

## 3. Results

### 3.1. The Results of the Initial MR Analysis

In the course of the MR analysis, we identified 249 SNPs with a strong association with TG, and these were subsequently extracted from the SNHL GWAS dataset. However, it is important to note that only 224 of the initially identified SNPs were present in the SNHL GWAS dataset, and proxy SNPs were not sought. Screening revealed a single SNP, namely “rs11868959”, with a significant association with the exposure in the outcome data. Among these SNPs, 10, namely rs1047964, rs11118310, rs1454687, rs154254, rs213484, rs2288912, rs28752924, rs5112, rs62112763, and rs7639927 were excluded due to their classification as “palindromes” during data harmonization. Additionally, we conducted MR-PRESSO and Radial MR analyses to identify outlier SNPs. No outlier SNPs were detected in the MR-PRESSO analysis, while 14 outlier SNPs, specifically rs10210970, rs11654777, rs17184382, rs2068888, rs2267373, rs28383314, rs28419182, rs28471687, rs6093446, rs684773, rs6913325, rs707931, rs76895963, and rs964184 (Figure 3A), were identified in the Radial MR analysis. In our study, we opted for SNPs with F-statistics exceeding 10 to mitigate potential bias associated with weak IVs. In the final analysis, a total of 200 SNPs were incorporated into the MR investigation. Detailed information on these SNPs is available in Appendix A.

To evaluate the causal association between TG and SNHL, we primarily employed the IVW method, complemented by MR-Egger, weighted median, and weighted mode analyses. The IVW results indicated a positive relationship between TG and SNHL, yielding an odds ratio (OR) of 1.14 (95% CI = 1.01–1.23, *p* < 0.001) (Figure 3B, Figure 4 and Appendix A). The Q and Q_P_ values of IVW and MR-Egger were 162.98 (0.971) and 162.63 (0.969), respectively, with Q_P_ > 0.05, indicating no heterogeneity. The Egger-intercept yielded a value of 0.001, in proximity to 0, with a standard error (SE) of 0.0017, and a *p*-value of 0.554, suggesting the absence of horizontal pleiotropy. Following the exclusion of outlier SNPs, the MR-PRESSO results indicated the absence of horizontal pleiotropy in this MR analysis (RSSobs = 164.68, *p* = 0.965). Table 1 provides an overview of the outcomes from the pleiotropy and heterogeneity assessments. Furthermore, the leave-one-out method was employed to visualize the IVW results. After sequentially removing individual SNPs, the IVW effect values of the remaining SNPs exhibited minimal fluctuations. This observation indicates the absence of highly influential SNPs among the IVs, reinforcing the stability and reliability of the results, as depicted in Appendix A.

### 3.2. Repeatable Verification of MR Results

To corroborate the credibility of this causal relationship, we conducted a replicated MR analysis. The GWAS dataset for TG was obtained from two separate European sources. Despite exhaustive attempts to access SNHL GWAS data from multiple sources, we were unable to do so, and thus resorted to employing the same SNHL GWAS dataset utilized in this study. We took care to adjust all analyses using the Bonferroni method, with statistical significance set at *p* < 0.025.

We screened 313 SNPs closely related to TG and extracted 289 SNPs in SNHL. We did not look for surrogate SNPs for complementation. SNP rs9373056 was found to be closely related to SNHL and was eliminated. Twelve palindromic sequence SNPs were subsequently eliminated (rs10822163, rs11274835, rs11722924, rs1316753, rs35140741, rs5112, rs551243, rs5755799, rs62112763, rs6752845, rs6916318 and rs9368503). Additionally, we conducted MR-PRESSO and Radial MR analyses to identify outlier SNPs. No outlier SNPs were detected in the MR-PRESSO analysis, while 16 outlier SNPs were identified in the Radial MR analysis (rs10210970, rs10883026, rs12504746, rs13269725, rs17184382, rs2068888, rs2187114, rs2267373, rs2604568, rs28383314, rs28752924, rs60856912, rs62427982, rs6532798, rs684773, and rs7215055) (Figure 5A). In our study, we opted for SNPs with F-statistics exceeding 10 to mitigate potential bias associated with weak IVs. In the final analysis, a total of 260 SNPs were incorporated into the MR investigation. Detailed information on these SNPs is available in Appendix A.

To evaluate the causal association between TG and SNHL, we primarily employed the IVW method, complemented by MR-Egger, weighted median, and weighted mode analyses. The IVW results indicated a positive relationship between TG and SNHL, yielding an OR of 1.09 (95% CI = 1.03–1.16, *p* < 0.001) (Figure 4, Figure 5B and Appendix A). The Q and Q_P_ values of IVW and MR-Egger were 194.13 (0.999) and 193.40 (0.999), respectively, with Q_P_ > 0.05, indicating no heterogeneity. The Egger-intercept yielded a value of 0.0012, in proximity to 0, with a SE of 0.0014, and a *p*-value of 0.391, suggesting the absence of horizontal pleiotropy. Following the exclusion of outlier SNPs, the MR-PRESSO results indicated the absence of horizontal pleiotropy in this MR analysis (RSSobs = 196.42, *p* = 0.999). Table 1 provides an overview of the outcomes from the pleiotropy and heterogeneity assessments. Furthermore, the leave-one-out method was employed to visualize the IVW results. After sequentially removing individual SNPs, the IVW effect values of the remaining SNPs exhibited minimal fluctuations. This observation indicates the absence of highly influential SNPs among the IVs, reinforcing the stability and reliability of the results, as depicted in Appendix A.

### 3.3. Meta-Analysis of MR

We conducted a meta-analysis, combining the results of both the initial and replicated MR analyses, which still indicated a positive correlation between TG and SNHL, with an OR value of 1.11 (95% CI = 1.06–1.16, *p* < 0.05, Figure 6). This leads to the conclusion that TG is a significant risk factor for SNHL, warranting further research and investigation. However, it is important to note that we did not present the results of the reverse MR analysis in this study, which involved a two-sample MR with SNHL as the exposure and TG as the outcome (Appendix A). Based on our findings, it is plausible that SNHL may not be a risk factor for TG.

## 4. Discussion

In this study, a potential causal link between TG and the incidence of SNHL was identified through a two-sample MR analysis, offering a promising foundation for future research.

Genetic abnormalities often involve gene mutations or alterations in genetic material, which may lead to the occurrence of specific traits or diseases. For example, abnormalities in genes such as MYO6, TECTA, ACTG1, WFS1, DFNA15, POU4F3, KCNQ4, and EYA4 can result in autosomal dominant non-syndromic hearing loss [30]. Patients with a 22q11.2 deletion may exhibit abnormalities in the semicircular canals, ossicles, vestibular aqueduct, and vestibule, particularly hearing loss [31]. Mendelian randomization is an experimental design method aimed at eliminating the influence of external factors on outcomes, especially in genetic studies. Mendelian randomization can be utilized to investigate the impact of genetic abnormalities. By randomly assigning individuals or samples to different treatment groups, the effect of genetic abnormalities on experimental outcomes becomes easier to observe and understand. This method helps reveal how genetic abnormalities influence the occurrence mechanism of specific traits or diseases, thereby providing a deeper understanding to aid the prevention and treatment of related diseases [32].

The IVW method evaluated the impact of each SNP on the outcome, and a meta-analysis was conducted using the outcomes as weights to assess the collective causal effect [32]. In this study, the IVW method operated under the assumption of no pleiotropy among these SNPs. Considering that the majority of the GWAS results were obtained following phenotypic standardization, it was presumed that a positive relationship existed between the outcomes and the exposure. Moreover, the statistical efficiency of the IVW method was considered the most informative [25]. Hence, in this study, IVW analysis was employed as the primary method for MR analysis, with MR-Egger, weighted median, and weighted mode analyses serving as supplementary tools to enhance the reliability of the findings. This study revealed a direct relationship between increasing TG levels and an elevated risk of developing SNHL, with an odds ratio (OR) of 1.14 (95% CI = 1.01–1.23). This association was consistently observed in the independently validated GWAS data, where the OR was 1.09 (95% CI = 1.03–1.16). Despite ample clinical evidence to the contrary, TG levels are not typically regarded as a primary risk factor for cardiovascular disease (CVD). Additionally, the AACE Guidelines do not definitively indicate the benefits of lowering TG levels to prevent CVD [33]. Nevertheless, taking into account that isolated TG elevation can effectively prompt changes in cholesterol transport and clearance mechanisms, disruptions in these processes may potentially play a role in impairing endothelial function [34]. Consequently, TG enrichment could also result in disturbances in cochlear microcirculation. Elevated TG levels are significantly correlated with the prevalence of SSNHL and its prognosis, suggesting that vascular damage may play an important role in the pathogenesis of SSNHL [35]. However, Saba ES et al. found that TG is not different in SSNHL compared to normal and that TG is not a causative factor for SSNHL [36]. The disparities in the outcomes of the aforementioned studies can be attributed to factors such as sample size, heterogeneity in demographic characteristics, and selection bias. In this study, we analyzed the potential link between TG and SNHL through MR, effectively circumventing the confounding influences mentioned above and enhancing the informativeness of the results.

If there is heterogeneity in the findings, it indicates that the analyses were confounded by confounding factors that affected the accuracy of the data [37]. The findings of this study revealed the absence of heterogeneity in both SNHL IVW analyses (Cochran Q = 162.98, *p* = 0.971) and MR-Egger regression analyses (Cochran Q = 162.63, *p* = 0.969). Similarly, the validation of SNHL IVW analyses (Cochran Q = 194.13, *p* = 0.999) and MR-Egger regression analyses (Cochran Q = 193.40, *p* = 0.999) also indicated no heterogeneity. It is demonstrated that the results of this study are less likely to be interfered with by other confounding factors and that the results of the MR analyses are reliable. Furthermore, the fundamental principle of MR analysis hinges on the notion that IVs exclusively influence the outcomes through the exposure in question. If IVs are capable of bypassing the exposure and directly affecting the outcomes, the results of such an MR analysis become devoid of meaning [21]. The validity of this principle can be ascertained by examining the presence of pleiotropy in the research findings. Thus, this study undertook pleiotropy testing using the MR Egger regression intercept. The results indicated the absence of pleiotropy in the analysis of SNHL (Intercept = 0.001, *p* = 0.554) and the repeated SNHL (Intercept = 0.0012, *p* = 0.391), further substantiating the effectiveness and reliability of the study’s results.

With the ongoing advancements in scientific research, some scholars have initiated investigations into the correlation between TG and SNHL. However, the connection between the two remains a subject of controversy, and the potential link between TG and the onset of SNHL is yet to be definitively established. Simultaneously, there is an absence of large-scale randomized controlled studies that systematically assess and compare the causal relationship between TG and the development of SNHL. In this study, we analyzed the potential causal relationship between TG and the onset of SNHL using Mendelian randomization (MR) to provide insights for future clinical endeavors. The findings indicate a conceivable connection between TG and the development of SNHL, suggesting that TG might play a role in the development of SNHL. Employees with hypertriglyceridemia face an elevated risk of noise-induced hearing loss. In workplaces characterized by high noise levels, alongside hearing protection programs, it is advisable to pay attention to workers’ serum levels and promote a healthy diet to help preserve their hearing thresholds [38]. TG enrichment may lead to cochlear microcirculation disorders and inadequate blood supply, resulting in hair cell damage and consequent hearing loss, suggesting that vascular damage may play an important role in the pathogenesis of SNHL [35]. Therefore, SNHL patients should be encouraged to undergo regular TG specialist follow-up to closely monitor the development of hypertriglyceridemia. In addition, early screening for TG to predict the risk of developing SNHL is recommended to favor early diagnosis and early treatment in more asymptomatic patients with TG.

Not all exposure factors have known strong genetic variants associated with them, limiting the applicability of MR methods. MR analysis is based on the assumption that genetic variants affect health outcomes solely through specific exposure factors, while in reality, genetic variants may influence health outcomes through multiple pathways, potentially introducing bias. Large-scale GWAS data are required for MR studies, and the quality and availability of this data may impact the accuracy and feasibility of the research. Despite MR’s ability to mitigate confounding effects, careful consideration and control of potential unmeasured confounders is still necessary. Due to the observational nature of MR studies, even if results support a causal relationship, alternative explanations cannot be entirely ruled out, such as the presence of unknown third factors or genetic variants correlated with other influencing factors [39,40]. Further research is needed to validate the viewpoint of this study, exploring the exact role of TG in the mechanism of sensorineural hearing loss (SNHL) through molecular biology and genomics methods, dissecting its interaction with cochlear cells or auditory neurons. Additionally, large-scale epidemiological studies will evaluate the association between TG levels and the incidence of hearing loss, providing a basis for preventive measures.

## 5. Conclusions

In conclusion, in this study, it was found that TG may promote the development of SNHL. This suggests that patients with SNHL should pay attention to changes in TG levels and avoid a high-fat diet. In addition, the exact mechanism between TG and SNHL is not clear, and more studies will help to explore this mechanism.

## Figures and Tables

**Figure 1 bioengineering-11-00438-f001:**
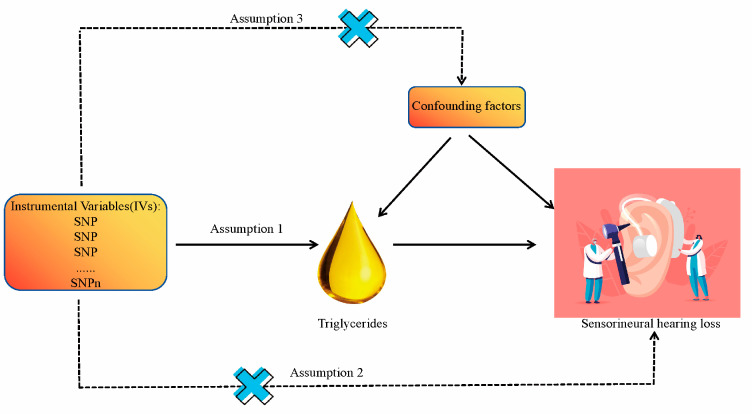
Research design overview and hypotheses for Mendelian randomization design.

**Figure 2 bioengineering-11-00438-f002:**
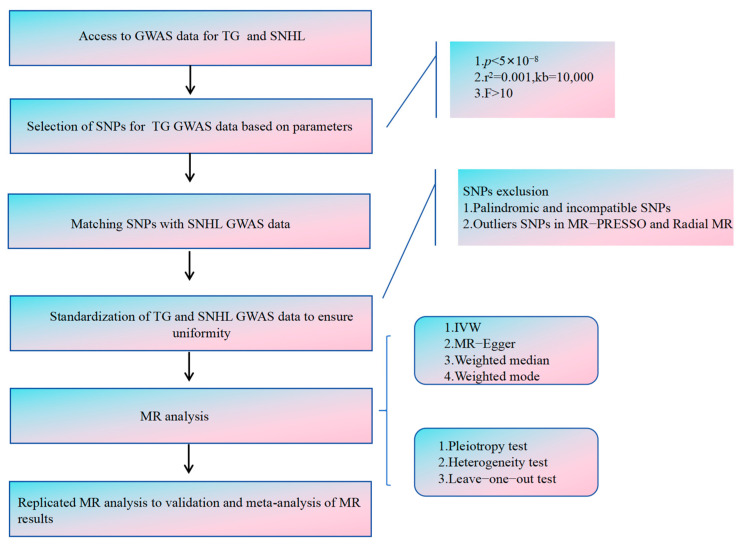
The flowchart illustrating the Mendelian randomization study aimed at elucidating the causal connection between TG and SNHL.

**Figure 3 bioengineering-11-00438-f003:**
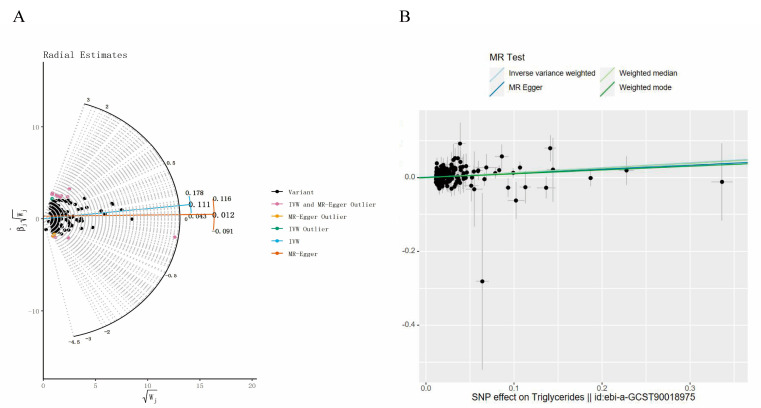
Related results in the replicated MR analysis: (**A**) Outlier SNPs identified by MR Radial; (**B**) Scatter plot depicting the MR results for TG and SNHL.

**Figure 4 bioengineering-11-00438-f004:**
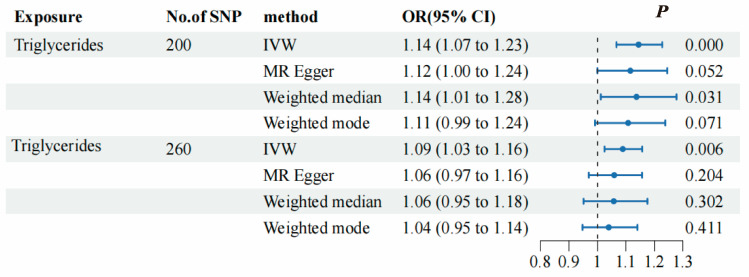
Forest plots presenting the causal relationship between TG and SNHL in both the initial MR analysis and the replicated MR analysis. The black dashed line corresponds to an OR value = 1, indicating that the exposure factor is not associated with the disease. When the OR value is >1, it indicates that the exposure increases the risk of the disease, i.e., it is a risk factor for the disease. When the OR value is <1, it means that the exposure decreases the risk of the disease, i.e., the exposure factor is protective against the disease.

**Figure 5 bioengineering-11-00438-f005:**
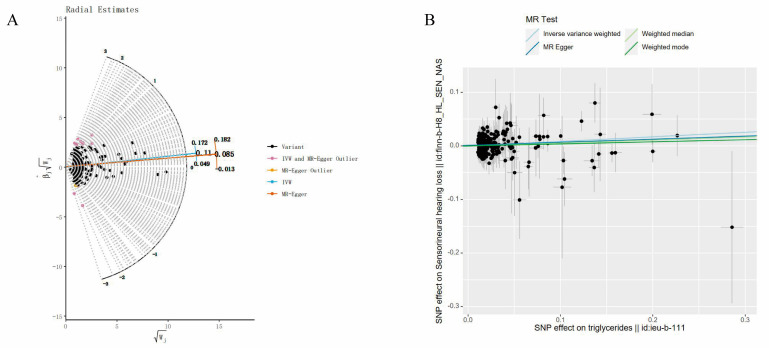
Related results in the replicated MR analysis: (**A**) Outlier SNPs identified by MR Radial; (**B**) Scatter plot depicting the MR results for TG and SNHL.

**Figure 6 bioengineering-11-00438-f006:**
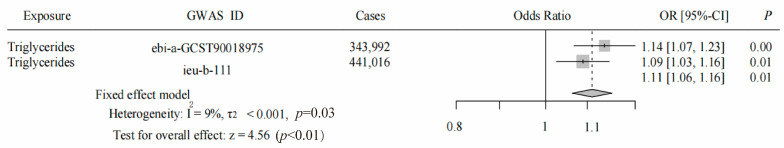
Meta-analysis results for TG and SNHL in both the initial and replicated MR analyses. The black dashed line represents the point estimate of the effect size of the summary result.

**Table 1 bioengineering-11-00438-t001:** Analysis of heterogeneity and horizontal pleiotropy results in the initial and repeated MR studies of TG and SNHL.

**MR Analysis**	**Exposure**	**Outcome**	**Heterogeneity**		**Pleiotropy Test**	**MR–PRESSO**	
			Cochran’s Q test	Rucker’s Q test	Egger intercept	Distortion test	Global test
			*p* value	*p* value	*p* value		
			IVW	MR–egger	MR–egger	Outliers	*p* value
the initial MR analysis							
	Trigycerides	Sensorineural hearing loss	0.971	0.969	0.554	NA	0.965
the replictated MR analysis							
	Trigycerides	Sensorineural hearing loss	0.999	0.999	0.391	NA	0.999

Note: NA is not applicable.

## Data Availability

The original contributions presented in the study are included in the article/Appendix A; further inquiries can be directed to the corresponding authors.

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
