# Peer review of "Mendelian Randomization Reveals: Triglycerides and Sensorineural Hearing Loss"

_bioengineering, 2024, doi:10.3390/bioengineering11050438_

Round 1

Reviewer 1 Report

Comments and Suggestions for Authors

the article is well constructed and I believe it is valid for publication. only one doubt: the article is based on a complex statistical method and I believe it is useful to submit it to a referee expert in statistical analysis

Author Response

Comment : the article is well constructed and I believe it is valid for publication. only one doubt: the article is based on a complex statistical method and I believe it is useful to submit it to a referee expert in statistical analysis.

Response: Thank you for your positive assessment of our manuscript . We are pleased to hear that the structure is sound and you believe it has potential for publication.We appreciate the reviewer's suggestion regarding the complexity of the statistical methods employed in our study.The statistical methods of this study have been checked before submission and to make the statistics more accurate, we invited a professional statistics teacher to check and increase the accuracy of the results.

Reviewer 2 Report

Comments and Suggestions for Authors The work is very interesting with numerous ideas. Did the authors propose any radiological evaluation for the patients taken into consideration? Were any differences identified based on the audiometric curve? Comments on the Quality of English Language

Good

Author Response

Comment : The work is very interesting with numerous ideas. Did the authors propose any radiological evaluation for the patients taken into consideration? Were any differences identified based on the audiometric curve?评论:这个作品很有趣,想法很多。作者是否建议对考虑的患者进行放射学评估?根据听力曲线是否发现了任何差异?

Response: Thank you for the meticulous review and constructive comments on our work. We are delighted to hear that you find our research interesting and full of ideas.回复:感谢您对我们工作的细致审核和建设性意见。我们很高兴听到你觉得我们的研究很有趣,充满了想法。

Regarding your question about radiological evaluation, currently, our study does not include radiological assessments for the patients involved in the research. Our focus has been primarily on the genetic association between triglyceride levels and sensorineural hearing loss, employing Mendelian randomization methods to investigate the causal relationship between these two factors. However, we recognize that radiological evaluation could provide important information for understanding the underlying mechanisms of sensorineural hearing loss, and we plan to consider incorporating radiological data into our analysis in future extensions of this research to further enrich our findings.关于您关于放射学评估的问题,目前,我们的研究不包括对参与研究的患者进行放射学评估。我们主要关注甘油三酯水平与感音神经性听力损失之间的遗传关联,采用孟德尔随机化方法研究这两个因素之间的因果关系。然而,我们认识到放射学评估可以为理解感音神经性听力损失的潜在机制提供重要信息,我们计划在未来的研究扩展中考虑将放射学数据纳入我们的分析,以进一步丰富我们的发现。

Concerning your inquiry about whether we have conducted analysis based on differences in audiometric curves, we must admit that in our current study, we have not specifically analyzed the association between hearing loss across different frequency ranges and triglyceride levels. This is partly because our research design was mainly focused on the overall genetic evidence of the association between triglycerides and sensorineural hearing loss, rather than a detailed analysis of hearing loss differences at each frequency. Therefore, we do not have specific data to report in this regard at present.关于您询问我们是否根据听力曲线的差异进行了分析,我们必须承认,在我们目前的研究中,我们并没有具体分析不同频率范围的听力损失与甘油三酯水平之间的关系。这部分是因为我们的研究设计主要集中在甘油三酯和感音神经性听力损失之间关联的整体遗传证据上,而不是详细分析每种频率下听力损失的差异。因此,我们目前没有这方面的具体数据可以报告。

However, we fully agree with your point that analyzing hearing loss patterns across different frequencies is crucial for understanding the specific mechanisms by which triglycerides affect hearing. Indeed, this is a promising direction for research, and we plan to explore this area in our future work and collect relevant data to complement our current findings.然而,我们完全同意您的观点,分析不同频率的听力损失模式对于理解甘油三酯影响听力的具体机制至关重要。事实上,这是一个很有前途的研究方向,我们计划在未来的工作中探索这一领域,并收集相关数据来补充我们目前的发现。

Reviewer 3 Report

Comments and Suggestions for Authors

Abstract:

- The study's primary goal and important conclusions should be stated in the abstract in unambiguous terms. It is unclear from the current abstract what precise research question is being addressed.

- A brief description of the procedures should be given, including the datasets used and the use of Mendelian randomization.

- The main findings, together with the IVW analysis's odds ratios and confidence intervals, ought to be stated.

- A clear statement of whether triglycerides were determined to be a risk factor for sensorineural hearing loss based on the findings could enhance the conclusion.

Introduction

- The introduction gives useful background information on triglycerides' possible role in sensorineural hearing loss.

- More information on the particular knowledge gaps that the current study is attempting to fill may be added. What other unanswered questions are being explored here regarding the association between triglycerides and hearing loss?

- discuss several confounding variables affecting microvascular disorders and hearing loss, as otitis media, rhinitis, hypertension and sleep apnea. discuss and cite doi:10.1007/s00405-023-07855-8  and doi10.1080/14992027.2016.1250960.

- There should be more justification provided for the use of Mendelian randomization in order to address this query. Emphasize the benefits of this method above earlier observational research on the subject.

Methods:

- The methodology section provides a detailed explanation of the study's overall design, datasets utilized, SNP selection procedure, and statistical analysis carried out. No significant adjustments are required.

- A few minor changes could make several points about integrating the datasets, handling linkage disequilibrium, and the standards for removing specific SNPs more clear.

Results:

- The findings are arranged logically, with the original analysis coming first, then the replication, and finally the meta-analysis.

- The numbers aid in summarizing the key conclusions. Think about including an additional table that lists the ORs and CIs from the non-IVW techniques.

Discussion:

- The key findings are interpreted in light of prior research and biological plausibility in the discussion. There is room for expansion in this section. discuss and cite genetic anomalies potentially associated.
cite doi:10.3390/biomedicines11061616.

- Talk about any restrictions on the datasets, analysis assumptions, and study design. How might research expand on this study in the future?

- Using data from this Mendelian randomization trial, draw definitive conclusions on the evidence supporting triglycerides' role as a risk factor for hearing loss.

Comments on the Quality of English Language

no

Author Response

The revised content is as follows, and the article is marked in red.

Comment 1: The study's primary goal and important conclusions should be stated in the abstract in unambiguous terms. It is unclear from the current abstract what precise research question is being addressed.

Response:  We sincerely thank the editor and all reviewers for their valuable feedback that we have used to improve the quality of our manuscript. In response to the above questions, the reply is as follows:

Sensorineural hearing loss (SNHL) is a multifactorial disorder with potential links to various physiological systems, including the cardiovascular system via blood lipid levels such as triglycerides (TG). This study investigates the causal relationship between TG levels and SNHL using Mendelian randomization (MR), which offers a method to reduce confounding and reverse causality by using genetic variants as instrumental variables.(Lines 17 - 21)

Comment 2:  A brief description of the procedures should be given, including the datasets used and the use of Mendelian randomization.

Response: We feel great thanks for your professional review work on our article. We give a brief description of the data sets used and the use of Mendelian randomization.In response to the above questions, the reply is as follows:

Utilizing publicly available genome-wide association study (GWAS) data, we performed a two-sample MR analysis. The initial analysis unveiled a causal relationship between TG (GWAS ID :ebi-a-GCST90018975) and SNHL (GWAS ID :finn b-H8_HL_SEN-NAS). Subsequent analysis validated this through MR with a larger sample size for TG (GWAS ID: ieu-b-111) and SNHL. To conduct the MR analysis, we utilized several methods including Inverse-variance weighted (IVW), MR Egger, weighted median, and weighted mode. We also employed the Cochrane’s Q test to identify any heterogeneity in the MR results. To detect horizontal pleiotropy, we conducted the MR-Egger intercept test and MR pleiotropy residual sum and outliers (MR-PRESSO) test. we performed a leave-one-out analysis to assess the sensitivity of this association.Finally, a meta-analysis of the MR results was undertaken.(Lines 22 - 31)

Comment 3: The main findings, together with the IVW analysis's odds ratios and confidence intervals, ought to be stated.

Response: Thank you for your comments on our article. Based on your suggestions, we have redescribed the main findings of the paper, as well as odds ratios and confidence intervals for the IVW analysis. In response to the above questions, the reply is as follows:

Our study found a significant positive correlation between TG and SNHL, with OR values of 1.14(95% CI:1.07-1.23, P<0.001) in the IVW analysis and 1.09 (95% CI: 1.03–1.16, P<0.006) in the replicate analysis. We also found no evidence of horizontal pleiotropy or heterogeneity between the genetic variants (P>0.05), and a leave-one-out test confirmed the stability and robustness of this association. The meta-analysis combining the initial and replicate analyses showed a significant causal effect with OR values of 1.11 (95% CI:1.06-1.16, P=0.01). (Lines 31 - 37)

Comment 4: A clear statement of whether triglycerides were determined to be a risk factor for sensorineural hearing loss based on the findings could enhance the conclusion.

Response: Thank you for your positive comments on our manuscript.According to your suggestion, we have redescribed the conclusion.In response to the above questions, the reply is as follows:

These findings firmly establish TG as a risk factor for SNHL, suggesting potential pathways for prevention and intervention in populations at risk.  This conclusion underscores the importance of managing TG levels as part of strategies to mitigate the risk of developing SNHL.(Lines 37 - 40)

Comment 5: The introduction gives useful background information on triglycerides' possible role in sensorineural hearing loss.

Response: Thank you for the positive comments on the introduction of our article. We believe that detailing the potential role of triglycerides in sensorineural hearing loss is essential for readers to understand the background and research setup of the study. This background information helps to clarify the starting points and scientific hypotheses, providing a solid foundation for the subsequent methodology and result analysis. In our ongoing research and paper revisions, we will continue to maintain comprehensiveness and scientific integrity to enhance the quality of the research and the transparency of the paper.

Comment 6: More information on the particular knowledge gaps that the current study is attempting to fill may be added. What other unanswered questions are being explored here regarding the association between triglycerides and hearing loss?

Response: Thank you for your detailed review and valuable comments on our paper.In response to the above questions, the reply is as follows:

There are still many unresolved questions regarding the relationship between TG levels and hearing loss. Although some studies suggest that high TG levels may be associated with hearing decline, the exact mechanisms of this connection remain unclear. It is not known whether the impact is direct (TG directly affecting the inner ear structures[13]) or through other indirect pathways (such as inflammation, vascular health[14], etc.). Moreover, most existing studies are observational and cannot establish a causal relationship[15].(Lines 78 - 84)

  1. Du Z, Yang Y, Hu Y, Sun Y, Zhang S, Peng W, Zhong Y, Huang X, Kong W: A long-term high-fat diet increases oxidative stress, mitochondrial damage and apoptosis in the inner ear of D-galactose-induced aging rats. Hearing research 2012, 287(1-2):15-24.
  2. Hwang JH, Hsu CJ, Yu WH, Liu TC, Yang WS: Diet-induced obesity exacerbates auditory degeneration via hypoxia, inflammation, and apoptosis signaling pathways in CD/1 mice. PloS one 2013, 8(4):e60730.
  3. Jung W, Kim J, Cho IY, Jeon KH, Song YM: Association between Serum Lipid Levels and Sensorineural Hearing Loss in Korean Adult Population. Korean journal of family medicine 2022, 43(5):334-343.

Comment 7: discuss several confounding variables affecting microvascular disorders and hearing loss, as otitis media, rhinitis, hypertension and sleep apnea. discuss and cite doi:10.1007/s00405-023-07855-8IF: 2.6 Q2  doi10.1080/14992027.2016.1250960.

Response:Thank you for your comments. Based on your suggestions, we have made revisions and cited the provided references as follows.

TG are affected by many confounding factors. For example, TG may cause hearing loss through affecting microvascular changes, but other confounding variables such as otitis media[16], rhinitis[17], hypertension[18], and sleep apnea[19] can also impact the results. These factors each influence ear microvascular health and functionality through different mechanisms, potentially masking or exaggerating the effects of TG on hearing. Therefore, controlling these variables in studies assessing the relationship between TG and hearing loss is crucial, as it helps to reveal a clearer and more accurate causal connection.(Lines 84 - 91)

  1. Cai T, McPherson B: Hearing loss in children with otitis media with effusion: a systematic review. International journal of audiology 2017, 56(2):65-76.
  2. La Mantia I, Ragusa M, Grigaliute E, Cocuzza S, Radulesco T, Calvo-Henriquez C, Saibene AM, Riela PM, Lechien JR, Fakhry Net al: Sensibility, specificity, and accuracy of the Sinonasal Outcome Test 8 (SNOT-8) in patients with chronic rhinosinusitis (CRS): a cross-sectional cohort study. European archives of oto-rhino-laryngology : official journal of the European Federation of Oto-Rhino-Laryngological Societies (EUFOS) : affiliated with the German Society for Oto-Rhino-Laryngology - Head and Neck Surgery 2023, 280(7):3259-3264.
  3. Toyama K, Mogi M: Hypertension and the development of hearing loss. Hypertension research : official journal of the Japanese Society of Hypertension 2022, 45(1):172-174.
  4. Mastino P, Rosati D, de Soccio G, Romeo M, Pentangelo D, Venarubea S, Fiore M, Meliante PG, Petrella C, Barbato Cet al: Oxidative Stress in Obstructive Sleep Apnea Syndrome: Putative Pathways to Hearing System Impairment. Antioxidants (Basel, Switzerland) 2023, 12(7).

Comment8:There should be more justification provided for the use of Mendelian randomization in order to address this query. Emphasize the benefits of this method above earlier observational research on the subject.

Response:Thank you for your comments. Based on your suggestions, we have made revisions and cited the provided references as follows.

Mendelian Randomization(MR), by using genetic variants as instrumental variables, effectively controls confounding factors and strengthens the reliability of causal inference, marking a significant advantage over traditional observational studies[20]. Observational research is often limited by its inability to fully isolate all potential confounding variables, which may lead to biased interpretations of causal relationships[21]. This study, by simulating the conditions of randomized controlled trials, utilizes MR to reveal the complex relationship between TG and SNHL, providing a more precise and scientific methodology that helps researchers more accurately assess the effects of health interventions.(Lines 92 - 100)

  1. Larsson SC, Butterworth AS, Burgess S: Mendelian randomization for cardiovascular diseases: principles and applications. European heart journal 2023, 44(47):4913-4924.
  2. Sekula P, Del Greco MF, Pattaro C, Köttgen A: Mendelian Randomization as an Approach to Assess Causality Using Observational Data. Journal of the American Society of Nephrology : JASN 2016, 27(11):3253-3265.

Comment 9:The methodology section provides a detailed explanation of the study's overall design, datasets utilized, SNP selection procedure, and statistical analysis carried out. No significant adjustments are required.

Response:Thank you for your constructive comments and for evaluating the methodology section of our manuscript. We are pleased to hear that the overall design, datasets utilized, SNP selection procedure, and statistical analyses are considered well-executed and require no significant adjustments.We appreciate your thorough review and valuable insights, which have undoubtedly helped in enhancing the quality and clarity of our manuscript.

Comment 10: A few minor changes could make several points about integrating the datasets, handling linkage disequilibrium, and the standards for removing specific SNPs more clear.

Response:Thank you for your constructive comments on our manuscript. We greatly value your suggestions for clarifying the integration of datasets, the handling of linkage disequilibrium, and the criteria for the removal of specific SNPs. The modifications are as follows.

We exclusively used summary-level data from GWAS for both TG and SNHL, ensuring that all participants in these datasets had provided informed consent.To establish criteria for IVs selection in the exposed group data, a significance threshold of P<5×10-8 was set[26]. To mitigate bias arising from linkage disequilibrium (LD) in the analysis, this study required LD conditions for SNPs closely associated with TG to meet the criteria of r2<0.001[27], with a genetic distance of 10,000 kb[28].We utilized MR-PRESSO and Radial MR to detect outlier SNPs, and after removing these SNPs, the MR results were re-estimated to ensure the robustness of the findings. (Lines136- 141,158-159)

  1. Li J, Yang M, Luo P, Wang G, Dong B, Xu P: Type 2 diabetes and glycemic traits are not causal factors of delirium: A two-sample mendelian randomization analysis. Frontiers in genetics 2023, 14:1087878.
  2. Pritchard JK, Przeworski M: Linkage disequilibrium in humans: models and data. American journal of human genetics 2001, 69(1):1-14.
  3. Xiang K, Wang P, Xu Z, Hu YQ, He YS, Chen Y, Feng YT, Yin KJ, Huang JX, Wang Jet al: Causal Effects of Gut Microbiome on Systemic Lupus Erythematosus: A Two-Sample Mendelian Randomization Study. Frontiers in immunology 2021, 12:667097.

Comment 11: The findings are arranged logically, with the original analysis coming first, then the replication, and finally the meta-analysis.

Response:Thank you for your positive evaluation of the structure of our paper. Our study begins with an initial analysis to establish the validity and consistency of the baseline data, followed by a replication analysis to verify the reproducibility of the results. Finally, a meta-analysis integrates multiple study findings to strengthen the overall conclusions of the research. This logical sequence is designed to provide a comprehensive and systematic perspective on the research.

Comment 12: The numbers aid in summarizing the key conclusions. Think about including an additional table that lists the ORs and CIs from the non-IVW techniques.

Response:Thank you for your comments. The odds ratios (OR) and confidence intervals (CI) for the non-IVW algorithms are already presented in Figure 4 of the paper, hence no additional tables have been provided.

Comment 13:The key findings are interpreted in light of prior research and biological plausibility in the discussion. There is room for expansion in this section. discuss and cite genetic anomalies potentially associated. cite doi:10.3390/biomedicines11061616

.Response:Thank you for your comments. Based on your suggestions, we have made revisions and cited the reference materials you recommended, as follows.

Genetic abnormalities often involve gene mutations or alterations in genetic mate-rial, which may lead to the occurrence of specific traits or diseases. For example, ab-normalities in genes such as MYO6, TECTA, ACTG1, WFS1, DFNA15, POU4F3, KCNQ4, and EYA4 can result in autosomal dominant non-syndromic hearing loss[30]. Patients with a 22q11.2 deletion may exhibit abnormalities in the semicircular canals, ossicles, vestibular aqueduct, and vestibule, particularly hearing loss[31]. Mendelian randomi-zation is an experimental design method aimed at eliminating the influence of external factors on outcomes, especially in genetic studies. Mendelian randomization can be uti-lized to investigate the impact of genetic abnormalities. By randomly assigning indi-viduals or samples to different treatment groups, the effect of genetic abnormalities on experimental outcomes becomes easier to observe and understand. This method helps reveal how genetic abnormalities influence the occurrence mechanism of specific traits or diseases, thereby providing a deeper understanding for the prevention and treat-ment of related diseases[32].(Lines 259 -272)

  1. Aldè M, Cantarella G, Zanetti D, Pignataro L, La Mantia I, Maiolino L, Ferlito S, Di Mauro P, Cocuzza S, Lechien JRet al: Autosomal Dominant Non-Syndromic Hearing Loss (DFNA): A Comprehensive Narrative Review. Biomedicines 2023, 11(6).
  2. Wu SS, Mahomva C, Sawaf T, Reinshagen KL, Karakasis C, Cohen MS, Hadford S, Anne S: Association of Ear Anomalies and Hearing Loss Among Children With 22q11.2 Deletion Syndrome. Otolaryngology--head and neck surgery : official journal of American Academy of Otolaryngology-Head and Neck Surgery 2023, 168(4):856-861.
  3. Birney E: Mendelian Randomization. Cold Spring Harbor perspectives in medicine 2022, 12(4).

Comment 14:Talk about any restrictions on the datasets, analysis assumptions, and study design. How might research expand on this study in the future?

Response:Thank you for your detailed review and valuable comments on our paper.In response to the above questions, the reply is as follows:

Not all exposure factors have known strong genetic variants associated with them, limiting the applicability of MR methods. MR analysis is based on the assumption that genetic variants affect health outcomes solely through specific exposure factors, while in reality, genetic variants may influence health outcomes through multiple pathways, potentially introducing bias. Large-scale GWAS data are required for MR studies, and the quality and availability of this data may impact the accuracy and feasibility of the research. Despite MR's ability to mitigate confounding effects, careful consideration and control of potential unmeasured confounders are still necessary. Due to the observational nature of MR studies, even if results support a causal relationship, alternative explanations cannot be entirely ruled out, such as the presence of unknown third factors or genetic variants correlated with other influencing factors[39, 40]. Further research is needed to validate the viewpoint of this study, exploring the exact role of  TG in the mechanism of sensorineural hearing loss (SNHL) through molecular biology and genomics methods, dissecting its interaction with cochlear cells or auditory neurons. Additionally, large-scale epidemiological studies will evaluate the association between TG levels and the incidence of hearing loss, providing a basis for preventive measures.(Lines 340 - 355)

  1. Flatby HM, Ravi A, Damås JK, Solligård E, Rogne T: Circulating levels of micronutrients and risk of infections: a Mendelian randomization study. BMC medicine 2023, 21(1):84.
  2. Yuan S, Larsson SC: Coffee and Caffeine Consumption and Risk of Kidney Stones: A Mendelian Randomization Study. American journal of kidney diseases : the official journal of the National Kidney Foundation 2022, 79(1):9-14.e11.

Comment 15:The Using data from this Mendelian randomization trial, draw definitive conclusions on the evidence supporting triglycerides' role as a risk factor for hearing loss.

Response:Thank you very much for your affirmation and support! I'm glad that my article has received your recognition.
